# Grape, Pomegranate, Olive, and Tomato By-Products Fed to Dairy Ruminants Improve Milk Fatty Acid Profile without Depressing Milk Production

**DOI:** 10.3390/foods12040865

**Published:** 2023-02-17

**Authors:** Fabio Correddu, Maria Francesca Caratzu, Mondina Francesca Lunesu, Silvia Carta, Giuseppe Pulina, Anna Nudda

**Affiliations:** Dipartimento di Agraria, Animal Science Unit, University of Sassari, Viale Italia, 39, 07100 Sassari, Italy

**Keywords:** agro-industrial by-product, grape pomace, pomegranate, olive cake, tomato pomace, bioactive compound, dairy cow, dairy sheep, dairy goat

## Abstract

The continuous increase in the cost of feeds and the need to improve the sustainability of animal production require the identification of alternative feeds, such as those derived from the agro-industrial sector, that can be effectively used for animal nutrition. Since these by-products (BP) are sources of bioactive substances, especially polyphenols, they may play an important role as a new resource for improving the nutritional value of animal-derived products, being effective in the modulation of the biohydrogenation process in the rumen, and, hence, in the composition of milk fatty acids (FA). The main objective of this work was to evaluate if the inclusion of BP in the diets of dairy ruminants, as a partial replacement of concentrates, could improve the nutritional quality of dairy products without having negative effects on animal production traits. To meet this goal, we summarized the effects of widespread agro-industrial by-products such as grape pomace or grape marc, pomegranate, olive cake, and tomato pomace on milk production, milk composition, and FA profile in dairy cows, sheep, and goats. The results evidenced that substitution of part of the ratio ingredients, mainly concentrates, in general, does not affect milk production and its main components, but at the highest tested doses, it can depress the yield within the range of 10–12%. However, the general positive effect on milk FA profile was evident by using almost all BP at different tested doses. The inclusion of these BP in the ration, from 5% up to 40% of dry matter (DM), did not depress milk yield, fat, or protein production, demonstrating positive features in terms of both economic and environmental sustainability and the reduction of human–animal competition for food. The general improvement of the nutritional quality of milk fat related to the inclusion of these BP in dairy ruminant diets is an important advantage for the commercial promotion of dairy products resulting from the recycling of agro-industrial by-products.

## 1. Introduction

In the livestock industry, the marked volatility of the commodity prices for feeds, coupled with the need to improve the sustainability of animal production, requires the identification of alternative feedstuffs, such as those derived from the agro-industrial sector. Even though the inclusion of by-products (BP) in livestock diets is an old practice [1], the idea to give a new life to some agro-industrial BP appears to be one of the biggest challenges of the food industry, which contributes to minimizing waste production [2]. Moreover, the use of BP in animal feeding generally replaces starch and protein concentrates, reducing competition with human foods and contributing to making the global food system more equitable [1,3].

Although the relevant amount of agro-industrial BP is produced worldwide [4] (about 250 million tons), their use and valorization are still low in the animal feed industry, with only a part of the material being reused. This is particularly true for many BP originating from the fruits and vegetables industrial processing, whose valorization and reuse are hampered by different limitations. One of the main problems is related to the huge humidity of these materials, which can promote fermentation processes and make them unusable for animal feeding purposes. Technical requirements for preservation, such as drying or ensiling, often entail high costs and energy consumption, balancing the environmental sustainability of using BP as feed [5]. For this reason, and in relation to the problem of their transportation from the farmers to the feed industries, most of the BP are used at the local level by livestock farms that are close to the site of production of the BP. The huge variation in nutrient composition and seasonal availability of some BP represent other limitations [6,7].

The interest in the main use of agro-industrial BP for animal feed as opposed to their use for biogas has also increased in view of their chemical composition, which appears to be of interest especially due to the presence of bioactive compounds (i.e., polyphenols) whose content still remains high after the industrial process. Bioactive compounds natively present in most of the agro-industrial BP could exert many beneficial animal and human effects, which confer added value to these materials [8]. Some of the worldwide agro-industrial BP, such as grape pomace, olive cake, tomato pomace, and pomegranate, received particular attention for their nutraceutical properties [2] and medical and health benefits [9].

Grape BP (i.e., seed, skin, and pomace) are one of the most investigated, being derived from the wine industry worldwide [10]; among them, grape pomace, or marc, is the most studied and is characterized by a high content of polyphenols [9], which remains high after the industrial process [11].

The industrial processing of pomegranate BP (i.e., seeds, sprouts, peel) has gained popularity for the content of several bioactive compounds, especially in peel [12] (which accounts for about 50% of the weight of the whole fruit and that is usually discarded at the end of the process [13].

Olive BP (i.e., leaves, cakes, and stones) have also been investigated as potential nutraceutical carriers for their antioxidant activity [14] and for their role in the prevention of various diseases [15].

Tomato BP (i.e., tomato pomace, a mix of skin, seeds, and residual pulp) [10] are another class of bio-residuals of particular interest since they are derived from the industrial process of one of the most relevant crops (tomato), which is widespread throughout the world [16]. The tomato residues represent at least 4% of the fruit weight and are rather rich in a variety of bioactive compounds [17], especially lycopene, a carotenoid with health-beneficial effects [18], whose concentration of the most bioactive isomer (cis lycopene) increases after heat processing [19]. Since these BP are sources of bioactive substances, especially polyphenols, they may play an important role as a new resource for improving the nutritional value of animal-derived products.

In small ruminants, the carryover of polyphenols to milk has been established [20,21,22]. The carryover of polyphenols and their metabolites into milk can natively enhance the nutraceutical properties of milk and dairy products because of the potential antioxidant, anticarcinogenic [23], anti-inflammatory [24,25], and antimicrobial activities of polyphenols [26].

In addition, polyphenols are effective in modulating the biohydrogenation process in the rumen of dietary polyunsaturated fatty acids (PUFA), and, hence, in the composition of milk fatty acids. The unsaturated FA in the diet, mainly linoleic acid (LA; cis-9,cis-12 C18:2), alpha-linolenic acid (LNA, cis-9,cis-12,cis-15 C18:3), and oleic acid (OA; cis-9 C18:1), are converted to saturated FA (SFA) by rumen microorganisms through the biohydrogenation process. During the biohydrogenation of LA, first occurs the isomerization to rumenic acid (RA; cis-9,trans-11 CLA), which is subsequently hydrogenated to vaccenic acid (VA; trans-11 C18:1) and finally to stearic acid (C18:0). The LNA is mostly isomerized to conjugated linolenic acid (CLA; cis-9,trans-11,cis-15), which is then hydrogenated to VA, and finally to C18:0. The intermediates VA and RA can escape from the rumen and accumulate in the tissues and milk. The VA is also the precursor of RA in the mammary gland, due to the enzyme delta-9 desaturase that catalyzes the addition of a cis-9 double bond to the VA [27]. 

Research in human and animal models evidenced the large health-promoting effects of RA, which were effective against cancer [28,29], atherosclerosis [30,31], and inflammation [32]. The important role of RA in obesity management has been well documented [29,33,34]. Interestingly, the use of dairy products naturally enriched in RA has been reported to reduce human plasma levels of the endocannabinoid anandamide, the LDL-cholesterol [35,36], and some markers of inflammation [37,38,39]. Hypolipidemic effects in animal models have also been associated with VA ingestions [40,41,42,43]. In addition, the intake of VA from children in the early years of life showed evident benefits in decreasing the risk of eczema [44].

Specifically, the polyphenols in the ruminant diets act indirectly, either limiting or suppressing the growth of rumen microorganisms involved in the PUFA biohydrogenation [45,46]. The negative or beneficial effects of polyphenols on the biohydrogenation of dietary PUFA can vary with animal species, the composition of the basal diet, the source of the polyphenols, and the amount of their inclusion in the diet [45].

Despite several investigations on this subject, the effectiveness of BP-containing bioactive compounds in improving milk lipid quality is still disputed. This is likely a consequence of the type of BP, the dose tested, the wide diversity of active compounds, the composition of the basic diet, and the length of the experiments.

Therefore, the present study summarized the effects of widespread agro-industrial BP such as grape, olive, tomato, and pomegranate BP on milk production, milk composition, and FA profile in dairy cows, sheep, and goats. The main objective of this work was to evaluate if the inclusion of BP in the diets of dairy ruminants, as a partial replacement of concentrates, could improve the nutritional quality of milk fat without having negative effects on animal production traits.

## 2. Literature Review Methodology

A systematic literature search was conducted in 2022 in Google Scholar, Web of Science, and PubMed by using the following keywords: grape, olive, pomegranate, winery, by-products and milk yield, milk composition, and fatty acids. This research was repeated by including the terms sheep, goat, cow, and ruminant, as well as bovine, ovine, and caprine. The inclusion criteria, for each by-product, were the measurement of intake, the level of by-product supplemented, the measurement of milk yield, and the main components (fat and protein) and/or the fatty acid profile of milk. For grape by-products, the initial search yielded 18 articles, and among them, only 13 have been selected because they include relevant data of interest; most of the publications used grape pomace or marc. For olive by-products, we selected 15 articles; most of them used olive cake (*n* = 12), and some of them in ensiled, dry, or pomace form. For pomegranate, only 10 articles fit our inclusion criteria. A total of thirty-eight publications from these databases were used, considering all three ruminant species. The dataset was prepared by including the results of the control and experimental groups; the extent of variation of each experimental group from the control group was calculated and expressed as a percentage.

## 3. Chemical Composition of Agro-Industrial by-Products 

Table 1 reports the chemical composition of the considered BP in ruminants’ nutrition. The chemical compositions of these materials vary widely in terms of protein, fat, and fiber. However, all are sources of nutrients, and, in particular, tomato pomace is relatively rich in fat and protein, while olive waste is an interesting source of fat, and all the BP have a remarkable concentration of neutral detergent fiber (NDF). Another interesting aspect is represented by the presence of bioactive compounds known as polyphenols; however, the content and type of these compounds can vary markedly among the parts of the plant waste, the industrial processing, the agro-climatic condition, and the cultivar used. The highest total polyphenol contents are in grape pomace and pomegranate. The main phenolic compounds were gallic acid, catechin, and peonidin-3-p-coumaroylglucoside in grape by-product [8], naringenin and rutin in tomato pomace [8], oleuropein and verbascoside, followed by hydroxytyrosol in olive [46], and hydrolyzable tannins, among which punicalagin is predominant, in pomegranate [47]. 

The health-promoting effects of polyphenols have been largely evidenced in laboratory animal and human studies as antioxidant [25], anti-inflammatory [48], neuroprotective [49], antithrombotic [48], anticarcinogenic [23], and control of glycemia [50] and obesity [51]. 

**Table 1 foods-12-00865-t001:** Chemical composition of selected agro-industrial by-products.

	Grape		Pomegranate		Olive		Tomato	
	Mean	Min–Max	Mean	Min–Max	Mean	Min-Max	Mean	Min–Max
By-product from agro-industrial processes ^1^, %	22	20–25	55.3	49.0–67.0	42.5	35–60	5	4–7.5
Moisture, %	71.2	63.0–81.7	-		44.4	32.0–60.0	87.7	78.4–97.0
Chemical composition								
Dry matter, % as fed	92.0	88.6–95.0	94.0	91.2–96.1	85.7	77.5–94.7	92.7	90.3–95.2
Protein, % DM ^2^	12.0	8.8–16.3	8.9	3.6–15.4	8.0	7.2–9.2	17.2	15.7–19.1
Fat, % DM	6.6	3.9–11.4	4.3	0.6–13.4	10.4	5.4–16.5	9.3	6.2–11.0
NDF ^3^, % DM	42.4	37.4–53.9	40.4	20.8–68.0	61.7	58.4–64.1	58.4	55.2–61.6
ADF ^4^, % DM	36.6	30.6–49.2	29.2	15.1–49.0	51.0	45.9–54.4	48.5	46.2–50.7
Ash, % DM	7.7	2.1–12.1	3.8	2.4–5.4	9.6	3.7–13.6	4.1	43.1–4.8
Total polyphenol, g/kg DM	35.9	3–90	52.8	27.2–95.3	10.9	4.1–17.7	4.9	2.3–6.4
Main fatty acids (% on total FA)								
Palmitic acid	9.6	5.5–12.5	3.8	3.5–4.2	13.1	9.1–17.5	13.9	11.5–15.6
Stearic acid	4.3	2.8–4.9	2.4	1.9–2.8	5.5	2.1–13.1	4.2	3.1–4.9
Oleic acid	15.2	9.6–19.2	7.2	5.4–9.2	65.5	54.3–75.8	19.7	17.6–20.7
Linoleic acid	66.9	61.1–74.0	7.1	5.9–8.1	9.9	6.6–13.8	54.8	52.2–57.1
Alpha linolenic acid	1.4	0.2–3.7	0.2	0.1–0.4	0.9	0.3–1.46	2.9	2.6–3.2
Punicic acid	-		73.8	71.2–77.2	-		-	

^1^ By-product, % arising from agro-industrial process. Grape by-products, % of by-product and composition: [52,53,54,55,56,57,58,59,60,61,62,63,64,65,66,67,68,69,70,71], polyphenols: [72,73,74], fatty acid: [67,70,75,76,77]; pomegranate by products, % of by-product and composition: [72,78,79,80,81,82], polyphenols: [77,83,84]; fatty acid: [72,82,85]; olive by-products, % of by-product and composition: [1,59,86,87,88,89,90,91,92,93,94,95,96,97], polyphenols: [91,98], fatty acids: [91,99,100,101,102]; tomato by products, % of by-product and composition: [16,83,91,103,104,105,106]; polyphenols [72,83,91]; fatty acids [72,83,91]; ^2^ DM = dry matter; ^3^ NDF = neutral detergent fiber; ^4^ ADF = acid detergent fiber.

All BP are already sources of FA, especially LA for grape and tomato BP, OA acid for olive BP, and punicic acid (PnA, cis9,trans11,cis13 C18:3) for pomegranate BP.

## 4. Voluntary Intake, Digestibility, and Fermentation Parameters of Agro-Industrial by-Products

The inclusion of different kinds of BP in the diets of dairy cows, dairy sheep, and dairy goats did not give univocal results in terms of voluntary intake, DM and nutrient digestibility, and ruminal fermentation parameters. The wide variability should be associated with differences among species, experimental conditions, source of BP, and doses. In general, high doses of polyphenols reduce feed intake, as evidenced by the strong negative correlation between these variables (R2 = 0.81; [8]). Therefore, the use of BP could exert no effect or could depress voluntary intake, DM and CP digestibility, VFA, and ammonia concentrations.

The decrease in feed intake and digestibility is merely associated with the high polyphenols and lignin content that characterize some BP [8].

In dairy ewes, the high ADL content of grape marc and tomato pomace and their polyphenol content and profile could be the reason for the reduction in DMI [83]. Compared to ewes, goats can better tolerate high polyphenol-rich BP since some strategies against their toxicity have been developed, such as the presence of proline-rich proteins in the saliva or the greater ability of the saliva to bind polyphenols [8]. Indeed, the feed intake of dairy goats was not compromised by the inclusion of pomegranate seed pulp [72,107], pomegranate seed oil [85], and tomato pomace [72] in the diet.

In dairy cows, the effect of BP on voluntary feed intake seems to be weak, probably because the level of BP inclusion in their diet is moderate. A negligible number of studies based on the use of grape BP in the diet of dairy cows evidenced no effect on feed intake [71,76,108,109,110]. At the same time, different kinds of pomegranate BP such as pomegranate peel [111], pomegranate pulp silage [112], or olive BP (i.e., ensiled olive cake; [113] can be effectively included in the diet of dairy cows without having negative effects on feed intake. However, some BP can decrease the fiber and protein digestibility, likely due to the low nutritional quality of CP in BP, as in grape residue silage [76] or tomato pomace [83], and to the high polyphenol content that can bind proteins or the cell wall. The decrease in protein digestibility could also occur as a result of the Maillard reaction during the industrial processing of BP [90]. As a consequence, the ruminal degradation of protein [8,76] and therefore the production of rumen ammonia could be reduced, as documented in dairy cows fed red and white grape marc [110] or pomegranate peel [111]. In dairy sheep, rumen ammonia and VFA concentrations were not changed by supplementation of grape or tomato pomace [114]. In dairy goats, supplementation with tomato pomace [72], pomegranate seed oil [85], and pomegranate seed pulp [72] did not alter ruminal fermentation parameters. Thus, in analogy to what is observed for voluntary feed intake, goats have a better ability to use BP-rich tannins than dairy cows and dairy sheep. Polyphenols present in this by-product can interact with rumen microbiota, affecting the fermentation process, and then bypass the rumen undegraded or partially degraded. They can partly be hydrolyzed in the intestine and absorbed, or they can interact with the intestinal bacterial microflora to form other metabolites.

## 5. Effect of Grape by-Products on Milk Yield, Composition, and Fatty Acids

The inclusion of grape BP in the diet of dairy cows (Table 2) did not evidence any significant depressive effect, both on milk yield and fat and protein contents, if included in the dose ≤150 g/kg dry matter intake (DMI). However, at higher doses of inclusion of grape BP in the diet, the extent of the reduction in milk yield reaches the maximum value of 12.3%. At these high levels of inclusion (≥25% of the whole diet, on a DM basis), the high concentration of lignin in grape BP (>40% of DM) notably influenced the lignin content of the diets, which were more than double compared to the control diet [75]. High dietary content of lignin could have decreased milk production by reducing the intake of metabolizable energy [8,75]. In addition, high doses of grape BP provided high levels of polyphenols, especially condensed tannins that, at high doses, are well known for their antinutritional effect [8]. The high levels of grape BP also evidenced a depressive effect on milk fat concentration (−19.2%) in dairy cows [75], probably because of the higher concentration of ether extract in the diet containing BP compared to the control, which can have reduced fiber digestion and consequently the precursor used by the mammary gland for the synthesis of milk fat. A reduction in fiber digestion can also be caused by the presence of high levels of polyphenols in the diet [45].

Although there is limited data available on small ruminants, no significant depressive effects on milk yield and their main components have been evidenced (Table 2). The significant reduction in fat and protein content observed in dairy sheep is likely due to dilution effects due to the increase in milk yield (+16.5%) in the group supplemented with grape marc (GM) compared to the control [83].

The effects of the grape BP on the selected individual FA and groups of FA in milk are summarized in Table 3. The inclusion of grape BP in the diet of dairy cows appears to be a clear feeding strategy to improve the nutritional quality of milk fat, as evidenced by a reduction of SFA and an increase in MUFA and PUFA. Among PUFA, a positive effect of grape BP was highlighted on rumenic acid (RA; cis-9,trans-11 CLA) (mean, +111%) and linoleic acid (LA; cis-9,cis-12 C18:2) (mean, +123%) contents. Moreover, vaccenic acid (VA, trans-11 C18:1) content increased with the inclusion of grape BP, with a mean of +121%. The RA and VA are two of the main intermediate products arising from the biohydrogenation of LA contained in grape BP by rumen bacteria. Their increase in milk, evidenced the accumulation in the rumen of biohydrogenation intermediate, with consequent escape to the mammary gland [27].

## 6. Effect of Pomegranate By-Products on Milk Yield, Composition, and Fatty Acids

Table 4 Similarly, any effect on milk production traits in dairy sheep has been reported by the published paper [82]. This could be explained by the lack of pomegranate BP effect on DMI in both cows [112] and sheep [82]. Actually, in sheep [82], the level of pomegranate pulp inclusion was high (70 g/kg of DMI), as was the content of the tannins in the BP studied (61 g/kg DM), considering that a content of tannins greater than 50 g/kg DM commonly decreases the intake and animal performance.

In dairy goats, a non-significant change in milk yield was found, whereas a significant increase in milk fat content was detected (Table 4). This is likely partly explained by the different composition of the pomegranate by-product used in goats, which consisted of seed pulp, which, compared to pomegranate pulp used in cows and sheep, is characterized by a higher content of ether extract (13.4% vs 3.9%, respectively) and NDF (41.5% vs 27.7%, respectively). This suggests a concentration effect on milk fat because the reduction in milk yield, although not significant, is due to the concomitant effect on the rumen fermentation process of high dietary fat and fiber. High NDF and fat can decrease the production of propionate and increase the production of acetate, thus reducing the supply of glucogenic substrates and increasing the supply of lipogenic substrates to the mammary gland.

Pomegranate is one of the main natural sources of a specific polyunsaturated fatty acid (PUFA), the cis-9,trans-11,cis-13 octadecatrienoic acid, also called punicic acid (PnA; Table 1). The PnA has a wide array of biological properties, including antidiabetic [121], anti-inflammatory [122], and anticarcinogenic activity against various forms of cancer [123,124,125,126]. Positive activities of PnA in the prevention and treatment of neurodegenerative diseases, such as Alzheimer’s and Parkinson’s [127], and in the modulation of lipid metabolism by the reduction in lipid accumulation and lipid droplet size in adipocytes [128,129], have been evidenced. The effects of dietary inclusion of pomegranate BP on selected individuals and groups of milk FA are reported in Table 5. Supplementation with pomegranate BP of ruminant’s diet appears an effective means of directly increasing the content of PnA in milk fat, due to the partial escape of PnA from the rumen, before its biohydrogenation to stearic acid (SA; C18:0), and subsequent carryover in the milk fat. The PnA has been found to be metabolized to RA and then to VA via saturation processes in the rat liver, kidney, and small intestine [130]. The PnA is probably also involved in the formation of rumen biohydrogenation intermediates, as the addition of pomegranate oil also increased the concentrations of VA (+47%) and RA (+164%) in milk [85]. The pathway of rumen biohydrogenation of PnA is not known, but in vitro incubation of PnA with *Butyrivibrio fibrisolvens* resulted in the formation of VA [131]. The VA could be partly converted into cis-9,trans-11 CLA in the mammary gland by the Delta9 desaturase enzyme (or stearyl-CoA desaturase) that catalyzes the addition of a cis-9 double bond on the trans-11 C18:1. This could be the reason for the positive effect of pomegranate BP supplementation, rich in LA, on VA and RA in milk fat of cows (mean, +21% and +19%), sheep (mean, +40% and +86%), and goats (mean, +64% and +115%) (Table 5). Despite the low content of LNA in pomegranate BP, it is also effective to increase alpha-linolenic acid (LNA, cis9,cis12,cis15 C18:3) content in milk fat of all species, even if the extent of the mean increase is markedly higher in goats than in sheep and cows (+64% vs +8.5 and +12.5%, respectively), likely because of the higher rumen escape of LNA toward the mammary gland in the former. However, the rearrangement of double bonds along the fatty acid acyl chains by isomerase activities in the rumen should be considered.

The positive effect on milk quality of the animals fed pomegranate BP was evidenced by the reduction in SFA content and the enhancement of MUFA and PUFA contents.

## 7. Effect of Olive by-Products on Milk Yield, Composition, and Fatty Acids

The inclusion of olive BP in ruminant diets did not significantly modify the milk production and composition in dairy cows (Table 6), except positive effect on milk protein content when BP was used in the destoned form [100].

Similarly, in dairy sheep and goats, the use of olive BP did not significantly modify milk yield and its main components if used in doses below 300 g/kg of DMI [138]. Olive cake is particularly low in digestibility and energy content, and the reduction in milk yield and protein content, observed with high-dose supplementation, is consistent with the reduced dietary energy supply. The conspicuous reduction in milk yield with olive leaves [91] has been related to the higher ADL:NDF ratio (1.60 vs 0.09, respectively) and the noticeably higher content of phenols compared to the control diet (6.65% vs 0.67%, respectively). The olive leaves used in [118] did not determine changes in milk production and composition compared to the control diet both in sheep and goats, because the by-product had conveniently replaced the alfalfa hay and wheat straw of the control diet.

Olive BP had a significant influence on the proportions of the main FA in milk fat (Table 7). In all ruminant species, almost all olive BP increased the concentration of oleic acid in milk (OA, C18:1 cis9) due to the lipid profile of olive BP [101], except for olive leaves. Moreover, the milk content of OA with supplementation of olive BP could be partly due to the desaturation of dietary stearic acid (C18:0), occurring in the mammary gland by the delta9 desaturase enzyme. Olive BP have significantly enhanced the VA and RA concentrations, likely due to the modulation of the biohydrogenation of LA in the rumen. Because of their unique FA composition, which is characterized by a high amount of LNA (from 25 to 35% of total FA), olive leaves, which were used in the diet of small ruminants as a partial or total replacement of forage, significantly increased the content of LNA in sheep and goat milk (+160% and +283%, respectively). The response of sheep to olive leaf supplementation was different from that of goats, considering that the extent of the LNA increase was markedly higher and that of VA and RA was lower in goats than sheep. This suggests in goats a lowering of the rumen biohydrogenation process that reduces the accumulation of VA and a higher rumen escape of LNA toward the mammary compared to sheep. Possible factors for different responses can include changes in total intake and feeding behavior between the two species.

The high LNA content in olive leaves makes this by-product an attractive and cheap source of n-3 PUFA in animal diets to improve the FA profile of milk.

Concerning the groups of FA, univocal results were highlighted for MUFA content, which increased in all species supplemented by olive BP. However, in sheep milk, the extent of the MUFA increase was higher compared to cow milk (means, +32% vs 14%, respectively). This result could be partially explained by the higher level of inclusion of olive BP in the diet of sheep in some of the considered studies [99,138].

## 8. Effect of Tomato by-Products on Milk Yield, Composition, and Fatty Acids

The effect of the inclusion of tomato BP in ruminant diets is reported in Table 8. Compared to other BP, tomato pomace is still little studied as an animal feed ingredient and probably extremely perishable due to the high level of moisture after industrial processing (Table 1), which makes its utilization difficult. The inclusion of tomato BP in ruminant diets did not evidence a significant depressive effect on milk production traits (Table 8) of all dairy species, except the reduction observed in goats at the highest dose tested (600 g/kg of DMI) [141]. Despite the limited energy content because of the high fiber concentration, TP is also a source of protein, minerals, and antioxidant compounds, which could explain the successful replacement of other concentrates in ruminant diets. A significant reduction in fat and protein content with tomato BP supplementation has been reported in dairy sheep, despite the low inclusion level [83]. Some compounds, such as flavanone naringenin and flavonols (mainly quercetin and kaempferol), which accumulate almost specifically in the peel, have been reported in the literature to have potential antimicrobial activity [142], and therefore such activity of phenolic compounds on rumen microflora can interfere with animal energy intake when used in high doses.

Naringenin is one of the most abundant polyphenols in tomato BP [83]. This polyphenol evidenced beneficial effect on cardiovascular disease [146], for the treatment of different types of cancer [147] and received recent attention in the treatment of liver disease [148]. Naringenin has been recently reported to exert antimicrobial properties in humans, especially against Gram-positive bacteria, such as *S. aureus*, including antibiotic-resistant strains [149].

Regarding the FA profile, the trials that investigated the inclusion of tomato pomace in ruminant diets are limited to small ruminants and showed increases in the concentrations of VA and RA in both sheep and goats (Table 9). The concomitant reduction in LA observed could be the consequence of extensive biohydrogenation of LA to RA [91] (or to VA [72,140]. One exception is represented in Murciano-Granadina goats [145], where a significant increase of LA and LNA (+11.9% and +38.5%, respectively) compared to the control diet has been observed, without a significant effect on the concentration of VA and RA. This result may suggest a reduced biohydrogenation of the LA and LNA, resulting in a larger amount of these PUFA in the small intestine, thus being available for mammary gland uptake. The inconsistency in the results observed in trials carried out on Murciano-Granadina goats [140,145] can be ascribed to differences both in the supplemented doses and in the physical form of the tomato BP, being as feed blocks [145] compared to the silage form [140]. In general, the FA profile of milk from animals supplemented with tomato BP highlighted an overall decrease in SFA and an increase in MUFA and total PUFA but also a significant increase in the n6-to-n3 ratio.

## 9. Conclusions

This paper demonstrates that the use of widespread agro-industrial BP, such as grape pomace, pomegranate, olive cake, and tomato pomace, as a substitution for part of the ratio ingredients, mainly concentrates, in general, does not affect milk production or its main components, but at the higher doses tested, it can reduce the milk yield within the range of 10–12%. However, the general positive effect on the milk FA profile was evident by using almost all BP at different tested doses.

The inclusion of these BP in the ration, which ranged from 5% up to 40% of DM without depressing milk, fat, or protein production as a rule, demonstrates the substitution power of concentrates of these feeds in terms of both economic and environmental sustainability, as well as the reduction in human–animal competition for food.

Furthermore, the general improvement in the nutritional quality of milk fat is an important advantage for the commercial promotion of dairy products resulting from the use of these BP.

Finally, the possible animal health benefits of ingesting substances with a generally well-documented anti-inflammatory effect, which would be in addition to the benefits mentioned above, i.e., improvement in animal health and welfare, remain to be evaluated.

## Figures and Tables

**Table 2 foods-12-00865-t002:** Effect of grape by-products included in dairy ruminants’ diets on milk yield, fat%, and protein%. Data are reported as the proportional difference between the treatment group, at the respective level of inclusion, and the control group.

Species	Breeds	By-Product Inclusion g/kg of DMI ^1^	By-Product Form ^2^	Milk Traits			References
				Yield	Fat	Protein	
Dairy cows	Danish Red Holstein	0.2 *	GP	1.6	−0.7	**−1.1**	[115]
	Holstein	10.0	GSGPE	**10.2**	−2.7	0.6	[108]
	Holstein	50.0	EGP	0.0	0.0	0.5	[76]
	Friesian	51.0	GP	−2.8	4.3	−2.4	[68]
	Holstein	56.4	GP	−9.2	−5.9	2.7	[109]
	Holstein	75.0	EGP	0.0	−0.4	−2.5	[76]
	Friesian	95.7	GP	−6.7	−5.2	−1.5	[67]
	Holstein	100.0	EGP	2.1	−5.2	0.3	[76]
	-	150.0	GP	2.1	−12.1	−0.9	[116]
	Holstein Friesian	247.5	GP red	**−7.6**	−6.4	**−3.2**	[110]
	Holstein Friesian	247.5	GP white	**−10.7**	1.7	**−1.6**	[110]
	Holstein Friesian	268.6	GP	**−12.3**	−0.8	−2.5	[75]
	Holstein Friesian	273.7	GP	5.5	**−19.2**	−1.1	[75]
Dairy sheep	Sarda	118.3	GS	2.0	4.0	−1.5	[117]
	Friesian	196.0	GP	−1.1	4.0	−6.8	[118]
	Sarda	41.1	GP	**16.5**	**−12.8**	**−6.8**	[83]
	crossbreed	50 *	GP	−7.8	−4.1	2.4	[119]
Dairy goats	Alpine	188.0	GP	−1.3	2.3	−0.3	[118]

Bold values indicate significant differences (*p* < 0.05) compared to the control group; ^1^ DMI = dry matter intake; ^2^ GSGPE = grape seed and grape pomace extract; EGP = ensiled grape pomace; GP = grape pomace; GS = grape seed; * calculated/estimated by authors.

**Table 3 foods-12-00865-t003:** Effect of grape by-products included in dairy ruminants’ diets on milk content of oleic acid (OA), vaccenic acid (VA), rumenic acid (RA), linoleic acid (LA), and alpha-linolenic acid (LNA) and saturated (SFA), monounsaturated (MUFA), and polyunsaturated (PUFA) fatty acids. Data are reported as the proportional difference between the treatment group, at the respective level of inclusion, and the control group.

Species	Breeds	By-Product Inclusion g/kg of DMI ^1^	By-Product Form ^2^	Fatty Acids ^3^					Groups of Fatty Acids ^4^						References
				OA	VA	RA	LA	LNA	SFA	MUFA	PUFA	n3	n6	n6/n3	
Dairy cows	Friesian	51	GP	0.3	**53.1**	−13.6	**14.0**	2.2	-	-	-	-	-	-	[68]
	Friesian	96	GP	0.5	**37.1**	**22.2**	**13.4**	2.2	**−1.8**	2.5	**7.6**	-	-	-	[67]
	Holstein Friesian	248	red GP	7.1	**64.4**	**60.4**	**253.2**	**−6.7**	−2.0	**10.2**	**159.4**	-	-	-	[110]
	Holstein Friesian	248	white GP	2.0	−1.9	**−3.8**	**172.5**	**−10.0**	−0.25	1.2	**106.5**	-	-	-	[110]
	Holstein Friesian	269	GP	**43.4**	**66.0**	**72.7**	**65.8**	−13.2	**−9.3**	**33.4**	**45.8**	-	-	-	[75]
	Holstein Friesian	274	GP	**74.0**	**509.4**	**527.3**	**219.1**	**5.9**	**−24.9**	**78.4**	**202.0**	-	-	-	[75]
	Holstein	50	EGP	-	-	-	-	-	−3.4	4.0	4.2	2.3	7.1	5.1	[76]
	Holstein	75	EGP	-	-	-	-	-	−5.4	6.4	6.9	11.6	−2.8	−12.7	[76]
	Holstein	100	EGP	-	-	-	-	-	−4.6	4.7	8.3	4.7	13.2	8.5	[76]
Dairy sheep	Sarda	41	GP	8.9	11.8	13.9	**20.7**	0.0	−3.2	6.4	**13.2**	−2.3	**17.0**	**21.3**	[114]
	crossbreed	50 *	GP	8.5	291.1	9.1	33.5	−3.5	−3.8	**17.1**	21.8	-	-	-	[119]
	Sarda	118	GS	**33.6**	**190.3**	**150.7**	**73.7**	−23.0	**−15.2**	**42.5**	**56.7**	−26.5	**64.0**	**124.0**	[120]
	Friesian	196	GP	14.5	**41.8**	56.3	**19.3**	15.2	−10.8	14.2	**27.1**	-	-	-	[118]
Dairy goats	Alpine	188	GP	17.3	−9.9	50.5	26.6	−12.5	−4.5	10.4	25.0	-	-	-	[118]

Bold values indicate significant differences (*p* < 0.05) compared to the control group; ^1^ DMI = dry matter intake; ^2^ GP = grape pomace; GS =grape seed; EGP = ensiled grape pomace; ^3^ OA = oleic acid (cis-9 C18:1); VA = vaccenic acid (trans-11 C18:1); RA = rumenic acid (cis-9,trans-11 CLA); LA = linoleic acid (cis-9,cis-12 C18:2); LNA = alpha-linolenic acid (cis-9, cis-12, cis-15 C18:3); ^4^ SFA = saturated fatty acids; MUFA = monounsaturated fatty acids; PUFA= polyunsaturated fatty acids; n-3 = n-3 fatty acids; n-6 = n-6 fatty acids. The dash indicates that the data was not reported in the publication; * calculated/estimated by authors.

**Table 4 foods-12-00865-t004:** Effect of pomegranate by-products included in dairy ruminants’ diets on milk yield, fat%, protein%, and the fat-to-protein ratio. Data are reported as the proportional difference between the treatment group, at the respective level of inclusion, and the control group.

Species	Breeds	By-Product Inclusion g/kg of DMI ^1^	By-Product Form ^2^	Milk Traits			References
				Yield	Fat	Protein	
Dairy cows	Holstein	10	CPE	5.5	−10.8	0.0	[74]
	Holstein	20	CPE	2.7	−1.8	3.1	[74]
	Crossbred Friesian	30	PP	2.8	−1.9	0.9	[111]
	Crossbred Friesian	37	PP	3.5	−1.9	1.3	[111]
	Holstein	40	CPE	8.2	−8.3	2.7	[111]
	Holstein	75	PS	0.4	−5.4	−2.3	[112]
	Holstein	87	PP	0.8	−2.8	−1.3	[73]
	Holstein	87	PP	3.0	2.0	−0.7	[73]
	Holstein	150	PS	−1.7	1.5	−3.6	[112]
Dairy sheep	Comisana	70	PP	2.3	0.1	−1.2	[82]
Dairy goats	Mahabadi	25	PSO	6.2	**7.3**	0.0	[85]
	southern Khorasan crossbred	60	PSP	−6.5	**14.6**	−2.6	[107]
	southern Khorasan crossbred	120	PSP	−6.9	**17.1**	−5.3	[107]
	Saanen	120	PSP	−2.0	**10.8**	4.2	[72]

Bold values indicate significant differences (*p* < 0.05) compared to the control group; ^1^ DMI = dry matter intake; ^2^ CPE = concentrate pomegranate extract; PP = pomegranate pulp; PS = pomegranate pulp silage; PSO = pomegranate seed oil; PSP = pomegranate seed pulp.

**Table 5 foods-12-00865-t005:** Effect of pomegranate by-products included in dairy ruminants’ diets on milk content of oleic acid (OA), vaccenic acid (VA), rumenic acid (RA), linoleic acid (LA), alpha-linolenic acid (LNA) and saturated (SFA), monounsaturated (MUFA), and polyunsaturated (PUFA) fatty acids. Data are reported as the proportional difference between the treatment group, at the respective level of inclusion, and the control group.

Species	Breeds	By-Product Inclusion g/kg of DMI ^1^	By-Product Form ^2^	Fatty Acids ^3^						Groups of Fatty Acids ^4^						
				OA	VA	RA	LA	LNA	PnA	SFA	MUFA	PUFA	n3	n6	n6/n3	References
Dairy cows	Holstein	75	PS	**4.3**	**18.8**	**14.9**	**11.7**	**12.5**	-	**−2.0**	**4.3**	**11.8**	**9.5**	**10.9**	**2.3**	[112]
	Holstein	150	PS	**10.4**	**22.9**	**23.9**	**9.3**	**12.5**	-	**−4.0**	**9.3**	**11.5**	**7.4**	**8.6**	**2.3**	[112]
	Holstein	87	PP	-	-	-	-	-		**−3.15**	**9.94**	**18.67**	-	-	-	[73]
	Holstein	87	PP	-	-	-	-	-		**−4.39**	**12.18**	**20.60**	-	-	-	[73]
Dairy sheep	Comisana	30	CPE	−2.1	-	-	12.5	3.4	-	-	-	-	-	-	-	[132]
	Assaf	70	PP	0.0	**39.4**	**85.5**	8.4	**13.5**	>1000 *	**−3.9**	1.7	**20.0**	**11.8**	6.8	−4.5	[82]
Dairy goats	Mahabadi	25	PSO	3.8	**47.0**	**162.9**	4.2	**121.2**	**312.5**	**−6.7**	**7.6**	**52.2**	**91.1**	8.9	−**44.8**	[85]
	southern Khorasan crossbred	60	PSP	−6.7	9.5	18.2	−9.6	1.3	206.7	-	-	-	-	-	-	[107]
	southern Khorasan crossbred	120	PSP	−4.1	**80.0**	**163.6**	−7.3	**75.7**	**693.3**	-	-	-	-	-	-	[107]
	Saanen	120	PSP	−0.9	**121.1**	-	−25.4	**55.3**	**745.5**	−1.1	3.6	**30.9**	-	-	-	[72]

Bold values indicate significant differences (*p* < 0.05) compared to the control group; ^1^ DMI = dry matter intake; ^2^ PS = pomegranate pulp silage; CPE = concentrate pomegranate extract; PP = pomegranate pulp; PSP = pomegranate seed pulp; PSO = pomegranate seed oil; ^3^ OA = oleic acid (cis-9 C18:1); VA = vaccenic acid (trans-11 C18:1); RA = rumenic acid (cis-9,trans-11 CLA); LA = linoleic acid (cis-9,cis-12 C18:2); LNA = alpha-linolenic acid (cis-9, cis-12, cis-15 C18:3); ^4^ SFA = saturated fatty acids; MUFA = monounsaturated fatty acids; PUFA = polyunsaturated fatty acids; n-3 = n-3 fatty acids; n6 = n-6 fatty acids. The dash indicates that the data was not reported in the publication; * not detected in control milk; 0.19 g/100 g of total FA.

**Table 6 foods-12-00865-t006:** Effect of olive by-products included in dairy ruminants’ diets on milk yield, fat%, protein%, and the fat-to-protein ratio. Data are reported as the proportional difference between the treatment group, at the respective level of inclusion, and the control group.

Species	Breeds	By-Product Inclusion g/kg of DMI ^1^	By-Product Form ^2^	Milk Traits			References
				Yield	Fat	Protein	
Dairy cows	Holstein	50	OP	−3.0	4.4	−1.2	[133]
	Holstein Friesian	100	DOP	-	7.3	**11.7**	[100]
	Holstein	100	OP	−3.8	6.1	0.0	[133]
	Holstein Friesian	100	OC	3.2	3.3	0.3	[102]
	Holstein Friesian	100	EOC	−1.4	4.6	2.8	[102]
	Holstein	130	OC	7.2	−16.0	2.6	[134]
	Holstein	150	OP	0.4	0.3	−2.6	[133]
Dairy sheep	Chios	72	EOC	−2.1	−2.1	-	[101]
	Chios	100	EOC	1.4	−2.0	1.6	[135]
	Chios	142	EOC	14.9	−4.2	-	[101]
	Chios	199	EOC	5.4	−3.6	−0.2	[135]
	Comisana	64	OC	**19.0**	1.4	−1.5	[136]
	Crossbreed	98	OC	-	10.0	0.0	[99]
	Awassi	200	OC	−3.5	−12.1	−2.0	[137]
	Crossbreed	244	OC	-	−5.0	−4.1	[99]
	Awassi	298	OC	**−8.0**	2.4	−4.0	[138]
	Awassi	300	OC	−5.8	0.1	−2.1	[138]
	Awassi	300	OL	**−18.0 ***	−5.5	2.1	[138]
	Friesian	751	OL	−1.33	−0.57	1.42	[118]
Dairy goats	Saanen	100	OCS	4.9	2.6	0.0	[139]
	Beni Arouss	200	OC	−12.8	-	-	[97]
	Saanen	200	OCS	−2.0	**20.5**	0.0	[139]
	Murciano-Granadina	202	OBSD	−8.7	**26.7**	9.6	[140]
	Alpine	702	OL	−2.13	1.17	1.27	[118]

Bold values indicate significant differences (*p* < 0.05) compared to the control group; ^1^ DMI = dry matter intake; ^2^ DOP = destoned olive pomace; OC = olive cake; OL = olive leaves; OP = olive pomace; EOC = ensiled olive cake; OBSD = olive by-product silage; OCS = olive cake silage; * Milk yield is expressed as energy corrected milk.

**Table 7 foods-12-00865-t007:** Effect of olive by-products included in dairy ruminants’ diets on milk content of oleic acid (OA), vaccenic acid (VA), rumenic acid (RA), linoleic acid (LA), alpha-linolenic acid (LNA), saturated (SFA), monounsaturated (MUFA), and polyunsaturated (PUFA) fatty acids. Data are reported as the proportional difference between the treatment group, at the respective level of inclusion, and the control group.

Species	Breeds	By-Product Inclusion g/kg of DMI ^1^	By-Product Form ^2^	Fatty Acids ^3^					Groups of Fatty Acids ^4^						References
				OA	VA	RA	LA	LNA	SFA	MUFA	PUFA	n3	n6	n6/n3	
Dairy cows	Holstein-Friesian	100	DOP	**17.6**	**64.2**	**21.7**	−0.5	4.8	**−4.0**	**14.2**	5.5	-	-	-	[100]
	Holstein-Friesian	100	OC	**21.0**	-	**10.2**	0.3	−26.6	**−4.9**	**14.6**	−0.3	-	-	-	[102]
	Holstein Friesian	100	EOC	**18.9**	-	4.0	−**11.8**	−35.9	−4.2	**13.2**	**−17.3**	-	-	-	[102]
Dairy sheep	Comisana	64	OC	-	-	-	4.1	−20.5	**−5.7**	**21.8**	−5	-	-	-	[136]
	Chios	72	EOC	-	-	22.2	2.5	−10.5	−7.1	30.3	3.2	-	-	-	[101]
	Crossbreed	98	OC	**31.0**	33.5	-	13.3	−8.3	**−8.1**	**26.5**	15	-	-	-	[99]
	Chios	100	EOC	**17.3**	18.7	**42.7**	**12.9**	−6.4	**−6.9**	**15.7**	**5**	-	-	-	[135]
	Chios	142	EOC	-	-	47.2	17.2	−8.8	−11.3	45.6	17.4	-	-	-	[101]
	Chios	199	EOC	**18.3**	14.2	**61.3**	**19.5**	−6.4	**−11.1**	**23.5**	**11.5**	-	-	-	[135]
	Crossbreed	244	OC	**55.6**	**66.5**	-	**42.0**	**−23.3**	**−14.9**	**48.2**	**22.3**	-	-	-	[99]
	Awassi	298	OC	**25.9**	0.0	2.7	**−18.1**	−5.3	−6.3	**22**	**−10.2**	−6.1	2.1	**13.4**	[138]
	Awassi	300	OC	**72.9**	−19.2	4.3	**−19.9**	−5.7	**−10.7**	**53.8**	−13.5	−5.6	**−16.0**	**−11.1**	[91]
	Friesian	751	OL	**30.3**	**24.9**	**71.1**	**121.9**	**121.7**	**−50.3**	**37.6**	**97.4**	-	-	-	[118]
Dairy goats	Saanen	100	OCS	9.3	-	-	−17.3	3.6	−2.2	11.1	−6.8	-	-	-	[139]
	Chios	142	EOC	-	-	47.2	17.2	−8.8	−11.3	45.6	17.4	-	-	-	[101]
	Alpine	702	OL	**34.4**	2.4	21.0	36.5	**282.9**	−6.5	23.1	**55.3**	-	-	-	[118]

Bold values indicate significant differences (*p* < 0.05) compared to the control group; ^1^ DMI = dry matter intake; ^2^ DOP = destoned olive pomace; OC = olive cake; OL = olive leaves; OP = olive pomace; EOC = ensiled olive cake; OBSD = olive by-product silage; OCS = olive cake silage; ^3^ OA = oleic acid (cis-9 C18:1); VA = vaccenic acid (trans-11 C18:1); RA = rumenic acid (cis-9,trans-11 CLA); LA = linoleic acid (cis-9,cis-12 C18:2); LNA = alpha-linolenic acid (cis-9, cis-12, cis-15 C18:3); ^4^ SFA = saturated fatty acids; MUFA = monounsaturated fatty acids; PUFA = polyunsaturated fatty acids; n-3 = n-3 fatty acids; n-6 = n-6 fatty acids. The dash indicates that the data was not reported in the publication.

**Table 8 foods-12-00865-t008:** Effect of tomato by-products included in dairy ruminants’ diets on milk yield, fat%, protein%, and the fat-to-protein ratio. Data are reported as the proportional difference between the treatment group, at the respective level of inclusion, and the control group.

Species	Breeds	By-Product Inclusion g/kg of DMI ^1^	By-Product Form ^2^	Milk Traits			References
				Yield	Fat	Protein	
Dairy cows	Holstein	72	TPS	−1.4	−7.8	0.3	[143]
	Holstein	100	TS	6.3	−2.7	0.0	[144]
Dairy sheep	Sarda	39	TP	−2.2	**−6.2**	**−6.5**	[83]
	Awassi	300	TP	−5.8	−2.4	−5.1	[91]
Dairy goats	Murciano-Granadina	134	TP	−7.4	1.1	−2.3	[145]
	Saanen	200	TP	**22.5**	18.1	3.4	[141]
	Murciano-Granadina	202	TS	−3.0	**13.1**	−3.6	[140]
	Saanen	240	TP	−2.0	6.0	3.5	[72]
	Saanen	400	TP	**28.3**	25.1	9.2	[141]
	Saanen	600	TP	**−13.3**	8.6	2.4	[141]

Bold values indicate significant differences (*p* < 0.05) compared to the control group; ^1^ DMI = dry matter intake; ^2^ TS = tomato silage; TP = tomato pomace; TPS = tomato pomace silage.

**Table 9 foods-12-00865-t009:** Effect of tomato by-products included in dairy ruminants’ diets on milk content of oleic acid (OA), vaccenic acid (VA), rumenic acid (RA), linoleic acid (LA), and alpha-linolenic acid (LNA), and saturated (SFA), monounsaturated (MUFA), and polyunsaturated (PUFA) fatty acids. Data are reported as the proportional difference between the treatment group, at the respective level of inclusion, and the control group.

Species	Breeds	By-Product Inclusion g/kg of DMI ^1^	By-Product Form ^2^	Fatty Acids ^3^					Groups of Fatty Acids ^4^						References
				OA	VA	RA	LA	LNA	SFA	MUFA	PUFA	n3	n6	n6/n3	
Dairy sheep	Awassi	300	TP	**92.2**	25.6	**76.5**	**−10.4**	17.0	**−15.0**	**68.0**	5.5	−1.9	7.4	9.5	[91]
	Sarda	39	TP	4.9	27.0	17.4	2.3	**−13.3**	−1.8	4.4	3.4	**−11.4**	2.9	**13.8**	[150]
Dairy goats	Murciano-Granadina	13	TP	1.3	8.5	8.8	**11.9**	**38.5**	−1.7	3.0	**14.7**	-	-	-	[145]
	Murciano-Granadina	202	TS	3.6	**188.4**	**32.0**	9.9	−12.2	**−4.6**	**12.4**	4.9	**−18.8**	4.0	**28.0**	[140]
	Saanen	240	TP	**22.9**	**100.0**	-	−24.0	5.3	**−5.1**	**23.8**	−3.2	-	-	-	[72]

Bold values indicate significant differences (*P*<0.05) compared to the control group; ^1^ DMI = dry matter intake; ^2^ TS = tomato silage; TP = tomato pomace; ^3^ OA = oleic acid (cis-9 C18:1); VA = vaccenic acid (trans-11 C18:1); RA = rumenic acid (cis-9,trans-11 CLA); LA = linoleic acid (cis-9,cis-12 C18:2); LNA = alpha linolenic acid (cis-9, cis-12, cis-15 C18:3); ^4^ SFA = saturated fatty acids; MUFA = monounsaturated fatty acids; PUFA = polyunsaturated fatty acids; n-3 = n-3 fatty acids; n6 = n-6 fatty acids. The dash indicates that the data were not reported in the publication.

## Data Availability

Data associated with the manuscript are available from the authors upon reasonable request.

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
