# Peer review of "Grape, Pomegranate, Olive, and Tomato By-Products Fed to Dairy Ruminants Improve Milk Fatty Acid Profile without Depressing Milk Production"

_foods, 2023, doi:10.3390/foods12040865_

Round 1

Reviewer 1 Report

Journal                  Foods (ISSN 2304-8158)

Manuscript ID                   foods-2198951

Type                         Review

Title                      Olive, grape, tomato, and pomegranate by-products fed to dairy ruminants improve milk fatty acid profile without depressing milk production

Dear authors:

This review presents some aspects of the use of vegetable waste in the feeding of farm animals.

The authors examined how the fatty acid composition of milk changes under the influence of the use of certain feeds.

The manuscript presents interesting research with an in-depth review of the scientific literature since 1971.

The title of the review is good and specific enough to keep the reader interested, so it's easy to understand.

The topic of the review is quite interesting and relevant at the present time. The co-authors tried to analyze the accumulated experience in the rational and economical use of industrial processing waste (olive, grape, tomato, and pomegranate by-products).

The review is quite logically structured and complies with the principles of presenting scientific information and research.

The structure of the review corresponds to the purpose of the study.

I think there are enough tables in the Review. The tables are clear.

I believe that the authors of the manuscript approached the analysis of the problem quite carefully and used a number of citations (146).

The co-authors of the review took it upon themselves to use the necessary number of sources of information in accordance with their structure of the manuscript and the plan of their own research.

Introduction

In order to improve the presented review, I recommend adjusting the stated purpose of the study in this section.

The main objective of this work was to evaluate if the inclusion of BP in the diets of dairy ruminants, as partially replacement of concentrates, can improve the nutritional quality of dairy products without negative effects on animal production traits.

The manuscript presents the results of studies of milk from farm animals, not dairy products.

The manuscript does not provide information on the breeds of cows, goats, and sheep.

Please add.

2. Chemical composition of agri-industrial by-products

Table 1. Chemical composition (mean ± SD) of some agri-industrial by-products

Authors are requested to provide a citation:

Protein, %

DM2 1

Fat, %DM

NDF3, % DM

ADF4, %DM

Ash, % DM

References

More than 40 percent of the sources of information presented in the Directory (bibliographic list) have been published over the past 5 years.

The authors carelessly formatted this section

Volumes not listed

Too many inaccuracies

I recommend that you carefully review this section in its entirety.

You need clarification on the design accepted in your journal

1 Molina-Alcaide, E.; Yáñez-Ruiz, D.R. Potential use of olive by-products in ruminant feeding: A review. Anim. Feed Sci. Technol. 2008, 147, 247–264.

11 Rockenbach, I.I.; Gonzaga, L.V.; Rizelio, V.M.; Gonçalves, A.E.D.S.S.; Genovese, M.I; Fett, R. Phenolic compounds and antioxidant activity of seed and skin extracts of red grape (Vitis vinifera and Vitis labrusca) pomace from Brazilian winemaking. food res. Int. 2011a, 44(4), 897–901.

Gonçalves, A.E.d.S.S.;

16 Alu'datt, M.H.; Alli, I.; Ereifej, K.; Alhamad, M. N.; Alsaad, A.; Rababeh, T. Optimization and characterization of various extraction conditions of phenolic compounds and antioxidant activity in olive seeds. Nat. Prod. Res. 2011, 25(9), 876–889.

https://www.tandfonline.com/action/showCitFormats?doi=10.1080%2F14786419.2010.489048

https://www.cabdirect.org/cabdirect/abstract/20113219688

12 Rockenbach, I.I.; Rodrigues, E.; Gonzaga, L.V.; Caliari, V.; Genovese, M.I.; Gonçalves, A.E.D.S.S.; Fett, R. Phenolic compounds content and antioxidant activity in pomace from selected red grapes (Vitis vinifera L. and Vitis labrusca L.) widely produced in Brazil. food chem. 2011B, 127, 174–179

50 Eleonora, N.; Dobrei, A.; Alina, D.; Bampidis, V.; Valeria, C. Grape pomace in sheep and dairy cows feeding. J. Hortic. for. 2014, 18, 146–150.

(correct=Nistor E., Dobrei A., Dobrei A., Bampidis V., Ciolac V = authors)

https://www.researchgate.net/publication/328812318_Grape_pomace_in_sheep_and_dairy_cows_feeding

56 Baldan, Y.; Riveros, M.; Fabani, M.P.; Rodriguez, R. Grape pomace powder valorization: a novel ingredient to improve the nutritional quality of gluten-free muffins. Biomass Converters. Biorefin. 2021, 1–13.

Volume number and page numbers not set by publisher

https://link.springer.com/article/10.1007/s13399-021-01829-8?utm_source=xmol&utm_medium=affiliate&utm_content=meta&utm_campaign=DDCN_1_GL01_metadata#citeas

https://doi.org/10.1007/s13399-021-01829-8

81 Ianni, A.; Di Maio, G.; Pittia, P.; Grotta, L.; Perpetuini, G.; Tofalo, R.; Cichelli, A.; Martino, G. Chemical–nutritional quality and oxidative stability of milk and dairy products obtained from Friesian cows fed with a dietary supplementation of dried grape pomace. J.Sci. Food Agric. 2019a, 99, 3635–3643

https://pubmed.ncbi.nlm.nih.gov/30629293/

88 Buffa, G.; Mangia, N.P.; Cesarani, A.; Licastro, D.; Sorbolini, S.; Pulina, G.; Nudda, A. Agroindustrial by-products from tomato, grape and myrtle given at low dosage to lactating dairy ewes: effects on rumen parameters and microbiota. ital. J. Anim. sci. 2020a, 19.1462–1471.

https://www.tandfonline.com/doi/full/10.1080/1828051X.2020.1848465

other

I recommend for publication the Review "Olive, grape, tomato, and pomegranate by-products fed to dairy ruminants improve milk fatty acid profile without depressing milk production" - foods-2198951 = after correction.

Thanks for giving me the opportunity to read your work. 

Prof. Dr. Maksim Rebezov

V. M. Gorbatov Federal Research Center for Food Systems of Russian Academy of Sciences, Moscow, Russian Federation

23.01.2023

E-mail: rebezov@ya.ru

Reviewer 2 Report

The authors present an interesting review which quality will be enhanced by including the strategy followed for the literature search by using the PRISMA method or any other valid quantitative method to assess the quality of the literature review. Please include a brief methodology section for the literature review (search terms, year range...). PRISMA is just an example of available methods. You can see in the article doi.org/10.3390/foods12030432 section about Literature Review Methodology. Please add a new section to your article.

Olive, grape, tomato, and pomegranate... please choose the order and follow it through the whole manuscript. In the discussion, the order is grape, pomegranate, olive, and tomato. If you choose this order then please change the order in the title, the order in the introduction, and the order in Table 1. The same order makes it easier for readers to follow.

 1.         Introduction

In general, my opinion is that the number of references is unnecessarily high. For instance, sentence in rows 54-55; I suggest that you focus only on studies regarding animal science. For instance, reference no 2 is not related to animals, but humans, which is not your focus. Data regarding the beneficial effects of reviewed BP in humans should not appear in your manuscript so much. Eliminating references that are not connected to animals will reduce the number of references in the reference list. You should comment on the effect on humans but only on what is found in milk (when ruminants eat selected BP and something is transferred to milk; or RA and VA benefits in humans).

Regarding the part of the introduction where you explain polyphenols and biohydrogenation in the rumen; you should elaborate a little bit more on how polyphenols influence biohydrogenation. Please read the manuscript doi.org/10.3168/jds.2018-14985 and include it in your reference list. Emphasize that polyphenols act indirectly on biohydrogenation. In paragraph rows 75-84 you don’t have any reference (the above-mentioned would fit just fine).

2.         Chemical composition of agri-industrial by-products

My opinion is that Table 1 and the accompanying text have many flaws. In the text you say fibre, but in table you presented ADF and NDF. You don’t have any comments regarding ADF and NDF content.

My suggestion is to present all results as a range (as you did for polyphenol content) and not as mean values that you calculated from the literature. You can write minimum and maximum that was reported in the literature and then maybe write in brackets the average value.

Regarding the FA that you listed in Table; the list is long and some of them are not important for your further discussion. I would remove myristic acid, palmitoleic acid, arachidic acid and gadoleic acid. They are not important and their content is rather small. Also, it would be more transparent for readers if you left only names (like palmitic acid) and delete number (e.g C16:0). A lot of numbers makes everything less transparent, and each name clearly tells what FA it is.

Header of Table 1; I would remove words pomace and cake.

3.         Effect of winery by-products on milk yield, composition, and fatty acids

In the subtitle you should use word “grape” and not winery (same comment to row 116).

Regarding Table 2; what is the significance of reporting F/C? Especially because you didn’t include it in the discussion. This parameter was calculated by you (you marked it with *) but I don’t understand the point.

Header of Table 2; MY should be replaced with the word Yield.

Below Table 2 you wrote “MY = milk yield, kg/day per head;”. However, the number in the table is a relative increase or decrease compared to the control group. The same comment applies to protein and fat content.

Regarding Table 3; VA and RA are very important and should be reported as they are. However, OA, LA and LNA should be replaced with SFA, MUFA, PUFA, n-6, n-3 and n-6/n-3 ratio. The reason why I am suggesting this is the fact that more beneficial profile of FA from a nutritional point of view refers to a lower proportion of SFA and a higher proportion of UFA, with the emphasis on n-3 PUFA, and a lower n-6/n-3 ratio. Therefore you should report and comment on that and not OA, LA and LNA individually.   

OTHER COMMENTS

Line 218: replace word ewe with sheep

Discussion regarding FA profile of milk in every section should be rewritten.

I suggest that you sort studies in all tables starting from the lowest inclusion level and finish with the highest inclusion level of analyzed BP. 

Reviewer 3 Report

This is a review paper that discusses the use of Olive, grape, tomato, and pomegranate by-products fed to ruminants on milk production and composition including fatty acid profile. The authors have covered on the milk production and composition aspects. However, utilization of these byproducts in ruminant diets needs consideration of feed intake, nutrient utilization, ruminal fermentation and production efficiency, which have not been covered in this review paper. Authors should consider to have great impact of this review papers instead of focusing only milk traits.

Some depressive effects on ilk yield in tables shown are not discussed well why there was reduction in milk yield and major milk components. This means authors need to well organize the review paper for the readers to clearly explain for them.

I find authors have nicely discussed for the fatty acid profile and I should expect on other variables also to improve the quality of this review.

Author Response

Referee 3

Regarding the first comments, authors have not attempted it. I would suggest to add at least one paragraph on each byproduct to give an overall overview on intake, digestibility and fermentation so that readers could also understand these effects on the utilization of the byproducts besides milk production and fatty acid composition. I believe it will not make the manuscript lengthy and will not be hard to handle. Moreover, I do not feel it is really a lengthy paper.

AU: Accepted. We added a paragraph with the effects of these BP on intake and digestibility.

Round 2

Reviewer 2 Report

Dear authors,

this review paper is is full of mistakes. You have many mistakes (especially in values). Every section needs to be rewritten and a detailed review of numbers in tables and text needs to be done.

Please see attached PDF file for specific comments.

Author Response

Answer to reviewer comments

Referee 2

this review paper is is full of mistakes. You have many mistakes (especially in values). Every section needs to be rewritten and a detailed review of numbers in tables and text needs to be done.

AU: With regret we read this suggestion that is unsupported by any evidence. We have double-checked all the calculations in the tables, and they are correct. We have, however, added the extent of variation of fatty acid groups to the tables of individual selected fatty acid. This shows, as evidenced by the half-empty columns, that there are very few scientific papers, that studied these by-products, which report the fatty acid groups.

R: Lines 48-49. This sentence refers to human? As I mentioned before, the focus of your paper are animals. This sentence is excess.

AU: The focus of papers is the use of BP in animals nutrition to improve the nutritional quality of foods (for human consumption) and we prepared this review for the journal “Foods”. As the target readership for this journal is broader than animal scientists, it is essential that references on papers referring to humans are also cited, otherwise this review would be meaningless. However, we rewrite and streamlined the sentence.

Lines 55-69. This comment refers to next 4 paragraphs (lines 52 to 69).

This section needs to be rewritten. Please exclude all references not related to animals. If you want to write about health benefits of selected BP than find references regarding animals. For instance, lycopene... does it have any impact on animals? If not then it is not relevant.

AU: As we report above, with the removal of the effects on humans, the value of the review is lost, which is aimed to study the effects of BP on the nutritional characteristics of milk fat (by reporting the  fatty acid profile), obviously for human nutrition. The title itself includes the purpose of this review, which summarises the role of these by-products in improving the nutritional characteristics of milk fat.

In this section you should focus on quantity of selected BP, are they for instance perishable, are they ease or hard to obtain (by farmers), is there any problem (for instance problem with transportation) or something like that. Find pros and cons for their usage in animal feeding.

The moisture content of tomato BP is very high (you reported 87,7% average), how it is reduced?

AU: Accepted. Although these aspects have been deeply investigated and reported by specific papers (the high humidity, the problem of transportation and so on are not the focus of this paper) we added a paragraph according to the reviewer suggestion. We prefer to make a brief and general dissertation rather than a punctual description of the problems related to each selected BP, because this is not the aim of our work.

R: Lines 73-86: when I read this section I have a feeling that you are going to report about a transfer of polyphenols to milk. And the fact is that the content of polyphenols in milk is not a part of your discussion. Therefore, this section is completely redundant.

AU: Accepted this part was removed from the manuscript.

Lines 87-96. This paragraph is the most important. Here you can add section about VA and RA, what are they, why are they important for humans.

AU: Accepted. The importance of VA and RA in humans have been moved here.

Also, in paragraph below you write about the improvement of the nutritional quality of milk. You should elaborate on that. What does it mean?

AU: We replaced” improvement of the nutritional quality of milk” with “improvement of the nutritional quality of milk fat”

As I mentioned before I think that it is necessary to write about n6, n3, SFA, MUFA, PUFA in milk.  In my opinion it is not acceptable to write about FA in milk without taking into account the bigger picture.

If you want to improve the nutritional quality of milk you want to decrease SFA, and increase UFA, introduction and discussion without that is unfinished!!!

AU: Accepted. We will add this information to the tables, when available in the cited papers. As we can see, many papers do not provide the family of fatty acids, so this review will also highlight this aspect. However, the bigger picture is too generic to explain the nutritional value of any foods, especially in light of the nutritional importance of certain specific fatty acids that fall into each group of FA. If I had to refer to fatty acid groups this review would never have been written due to lack of information in most published works.

Lines 99-100. Why do you write for instance only tomato pomace when you included tomato silage as well? I would delete words pomace and cake and write only BP (in general).

AU: Accepted. we changed specific by product to a general BP

Lines 123. the selected ones. I am sure there are other BP that are tested more.

AU: accepted. most tested was changed in “considered”

Lines 123-128. This sentence is to long. Please divide it shorter sentences.

AU: accepted, the sentence was shortened and corrected.

Lines 130. Polyphenol

AU: accepted

Lines 135-137. Again, I don't see the relevance

AU: As explained above, we believe that this kind of information is useful considering that the focus of the review (and the special issue) is milk as  food for human consumption.

Lines 139-140. linoleic, not linolenic

AU: Thanks for the comments. Accepted

Line 140. Latter in the text you use abbreviation OA. Please introduce it during first appearance in text and use abbreviation from that point forward.

AU: Thanks for the comments. Accepted

Line 140. Latter in the text you use abbreviation PA. Please introduce it during first appearance in text and use abbreviation from that point forward

AU: Thanks for the comments. Accepted

Table 1. The number of decimal places should be uniform. You use from 0 to 2 decimal places. Please uniform. I suggest that you report with 1 decimal.

AU: Thanks for the comments. Accepted and corrected accordingly.

Line 142. Selected

AU: Thanks for the comments. Accepted

Table 1. Dear authors,

a lot of effort, time and concentration is required to write this kind of table. I have found NUMEROUS mistakes. I will list only one:

tomato protein - you reported 18,1-21,0; reference no 4 is one of the references that you used (and this is yours previous review paper) and in that paper you reported that protein in tomato pomace is 15,7. This is not in line what you are reporting now.

PLEASE REVISE THIS WHOLE TABLE

AU: Thanks for the comments. Accepted. The table was completely revised. Values and relative references were checked and corrected. Data from studies reporting more than two values for the same parameters were averaged and then used for the calculation of the mean reported in the table.

Table 1. I think it is enough to write Chemical composition (without as feed ingredient, that is implied).

AU: Thanks for the comments. Accepted and corrected accordingly.

Table 1. You already used superscript 1, this one should be 2. Please correct also below table.

AU: Thanks for the comments. Accepted and corrected accordingly.

Table 1. ALA, You already used superscript 1, this one should be 2. Please correct also below table.

AU: Thanks for the comments. Accepted and corrected accordingly.

Table 1. Are all these references about cake?

AU: Thanks for the comments. References were about olive by products.

Lines 164. -19.2%

AU: Accepted

Lines 177-178. I do not see numbers +121, +111 and +123 in Table3??? Instead I see 509.4, 527.3 and 253.2.

AU: Accepted. The numbers reported in the text (+121, +111 and +123) refer to the mean of the values reported in table for the cows. We better clarified it in the text.

Lines 185. Why don't you use abbreviation RA?

AU: Thanks for the comments.Amended.

Table 3. Reference no 63 reported on milk FA profile. However they reported the content of n6, n3, n6/n3, MUFA, PUFA, SFA, in other words they reported FA profile of milk relevant for human health. That is why I insist that you include above listed parameters in your table and discussion. I do not mind if you leave OA, LA and ALA but please expand the list and discussion.

AU: Accepted. The authors completely agree with the reviewer that data on MUFA, PUFA, SFA, PUFAn6, PUFAn3, and their ratio (n6/n3) provide important information about the relationship between milk composition and human health. In the first round of revision, we preferred to not report such parameters because of their lack in many works; moreover, including individual FA may help to better understand the effect of each BP (particularly in relationship with its peculiar FA profile) on the milk FA profile.

In this revised version we added such information according to the reviewer suggestion.

Lines 242-244. In table 5 references that report the increase of PA in milk are 43, 44, 110 and 116. Why didn't you list them here?

AU: Thanks for the comments. Actually, the text did not refer to the table but to two specific works. However we changed the sentence and highlight the results reported in the table.

Lines 253-255. The numbers in brackets are again wrong!

AU: as explained above, the numbers reported in the text refer to the mean of the values reported in table for the cows. We better clarified it in the text.

Line 259. Again wrong numbers in brackets!

AU: as explained above, the numbers reported in the text refer to the mean of the values reported in table for the cows. We better clarified it in the text.

Table 5. Decide are you going to use abbreviation PA or PnA

AU: Accepted. We replace PA with PnA in all text

Line 291. when BP was used in destoned form

AU: Accepted.

Lines 297. Use abbreviation ADL

AU: Accepted.

Line 311. Are you using abbreviation ALA or LNA?

AU: Accepted and fixed

Line 311. Wrong number again!

AU: as explained above, the numbers reported in the text refer to the mean of the values reported in table

Line 320. This is not acceptable in science language

AU: Accepted and removed.

Lines 362-363. Is it found in animal milk as well? Human milk is not relevant to your research.

AU: accepted. The sentence has been deleted

Lines 404-415. This section doesn't belong to conclusion, rather introduction

AU: accepted and deleted

Reviewer 3 Report

Authors have revised the second commend and I believe the quality of the manuscript has improved further.

Regarding the first comments, authors have not attempted it. I would suggest to add at least one paragraph on each byproduct to give an overall overview on intake, digestibility and fermentation so that readers could also understand these effects on the utilization of the byproducts besides milk production and fatty acid composition. I believe it will not make the manuscript lengthy and will not be hard to handle. Moreover, I do not feel it is really a lengthy paper.

Author Response

(The authors gave the same response as above.)
